# Intra-aneurysmal pressure changes during stent-assisted coiling

**Piotr Piasecki, Piotr Ziecina, Krzysztof Brzozowski, Marek Wierzbicki, Jerzy Narloch** *

Department of Interventional Radiology, Military Institute of Medicine, Warsaw, Poland

* jerzy.narloch@gmail.com

**Data Availability Statement:** All relevant data are within the paper and its Supporting Information files.

**Funding:** The authors received no specific funding for this work.

## Abstract

We aimed to examine aneurysm hemodynamics with intra-saccular pressure measurement, and compare the effects of coiling, stenting and stent-assisted coiling in proximal segments of intracranial circulation. A cohort of 45 patients underwent elective endovascular coil embolization (with or without stent) for intracranial aneurysm at our department. Arterial pressure transducer was used for all measurements. It was attached to proximal end of the microcatheter. Measurements were taken in the parent artery before and after embolization, at the aneurysm dome before embolization, after stent implantation, and after embolization. Stent-assisted coiling was performed with 4 different stents: LVIS and LVIS Jr (Microvention, Tustin, CA, USA), Leo (Balt, Montmorency, France), Barrel VRD (Medtronic/ Covidien, Irvine, CA, USA). Presence of the stent showed significant reverse correlation with intra-aneurysmal pressure—both systolic and diastolic—after its implantation ($r = -0.70$ and $r = -0.75$, respectively), which was further supported by correlations with stent cell size—$r = 0.72$ and $r = 0.71$, respectively ($P<0.05$). Stent implantation resulted in significant decrease in diastolic intra-aneurysmal pressure ($p = 0.046$). Systolic or mean intra-aneurysmal pressure did not differ significantly. Embolization did not significantly change the intra-aneurysmal pressure in matched pairs, regardless of the use of stent ($p>0.05$). In conclusion, low-profile braided stents show a potential to divert blood flow, there was significant decrease in diastolic pressure after stent placement. Flow-diverting properties were related to stent porosity. Coiling does not significantly change the intra-aneurysmal pressure, regardless of packing density.

## Introduction

Endovascular coiling has an established position in treatment of intracranial aneurysms. [1] Self-expandable stents extended the indications for endovascular therapy for wide-necked and complex aneurysms, which were not amenable to coiling [2, 3]. Placement of the stents across the aneurysm neck provides the scaffold for the coils, stabilizing their position and preventing the protrusion into the parent artery. Stent struts cover the aneurysm orifice, and modify the inflow of blood into the sac. This effect is thought to be dependent on strut density and the degree of blood inflow impairment [4–6]. Recently, low-profile, self-expandable, braided

**Competing interests:** The authors have declared that no competing interests exist.

intracranial stents (LEO Baby (Balt, Montmorency, France) and LVIS Jr. (MicroVention, Tustin, California)) have been used in monotherapy of distally located small aneurysms [7, 8]. There is also a limited number of case series on the application of stent monotherapy in treatment of aneurysms located proximal to the circle of Willis [9–12]. Hemodynamic effects were assessed by angiography alone by computational fluid dynamics simulations (CFD) [7–12].

In this study, we aimed to examine aneurysm hemodynamics with intra-saccular pressure measurement, and compare the effects of coiling, stenting and stent-assisted coiling in proximal segments of intracranial circulation.

## Materials and methods

A cohort of 45 patients underwent elective endovascular coil embolization (with or without stent) for intracranial aneurysm at our department. Patient data regarding demographics, angiography and hemodynamics were evaluated retrospectively. Informed consent was obtained from all patients. The study was approved by Institutional Review Board—Bioethical Committee of Military Institute of Medicine (decision 43/WIM/2011).

Arterial pressure transducer was used for all measurements. It was attached to proximal end of the microcatheter. Before the measurement the transducer was zeroed at the level of the right atrium. The microcatheter was carefully positioned under fluoroscopic guidance to avoid adherence to or damage to the vessel wall; measurements were taken in the parent artery before and after embolization, at the aneurysm dome before embolization, after stent implantation, and after embolization. Measurements were recorded, as soon as the read values stabilized (i.e. average of 30 seconds). Simultaneous recording of systemic blood pressure and heart rate were taken. Stent-assisted coiling was performed with 4 different stents: LVIS and LVIS Jr (Microvention, Tustin, CA, USA), Leo (Balt, Montmorency, France), Barrel VRD (Medtronic/Covidien, Irvine, CA, USA).

During the procedure, all patients were under general anesthesia with mean systemic blood pressure maintained between 65 and 100 mmHg. Systemic heparinization was achieved by weight-adjusted intravenous bolus and maintenance dose of 1000 IU of unfractioned heparin/1 hour.

A combination of 8-Fr introducer sheath and a 6-Fr guiding catheter, or a 7 -Fr long sheath with a 5- Fr guiding catheter were used. A standard 1.7-Fr coiling microcatheter was used for aneurysm coil embolization and pressure monitoring (Headway 17 (Microvention, Tustin, CA, USA) or Echelon 10 (Medtronic, Dublin Ireland)). A stent implantation was performed with a dedicated 2 Fr microcatheter–it was not utilized for pressure monitoring.

Descriptive statistics of all variables are presented. Wilcoxon signed-rank test was used to compare differences between matched samples. For unmatched samples Mann-Whitney U test was used. Alpha<0.05 was considered significant. All relevant data are within the manuscript and its S1 File.

## Results

Intra-aneurysmal pressure measurements were performed in 45 patients (4 men and 41 women, aged 67.3±4.6 years and 58.2±13.6 years, respectively). Descriptive statistics regarding intra-aneurysmal pressure and aneurysm morphology were collected in Tables 1 and 2. For schematic morphological measurements please refer to Fig 1. Mean systemic blood pressure was 105.7±15.4 mmHg in systole and 62.7±10.5 mmHg in diastole; mean heart rate was 66±8/min. Locations of treated aneurysms were: (1) internal carotid artery (ICA)– 17 cases of right ICA and 17 cases of left ICA, total 76%; (2) basilar artery (BA)– 6 cases, 13%; (3) anterior cerebral artery (ACA)– 3 cases, 6%; and (4) middle cerebral artery (MCA)– 2 cases, 5%.

**Table 1. Descriptive statistics regarding intra-aneurysmal pressure in subgroups with and without stenting.**

| POSITION | VARIABLE | MEAN | SD |
|---|---|---|---|
| **WITH STENT** | | | |
| **parent artery** | systolic pressure (mmHg) | 72 | 12.7 |
| | diastolic pressure (mmHg) | 64 | 11.9 |
| **aneurysm dome before embolization** | systolic pressure (mmHg) | 71.3 | 8.6 |
| | diastolic pressure (mmHg) | 65.3 | 7.8 |
| **aneurysm dome after stent implantation** | systolic pressure (mmHg) | 68.3 | 15 |
| | diastolic pressure (mmHg) | 62.2 | 14.4 |
| **aneurysm dome after embolization** | systolic pressure (mmHg) | 72.1 | 12.5 |
| | diastolic pressure (mmHg) | 64.4 | 9.4 |
| **WITHOUT STENT** | | | |
| **parent artery** | systolic pressure (mmHg) | 75.1 | 18.2 |
| | diastolic pressure (mmHg) | 67.5 | 17.9 |
| **aneurysm dome before embolization** | systolic pressure (mmHg) | 75.5 | 18.4 |
| | diastolic pressure (mmHg) | 69.3 | 17.5 |
| **aneurysm dome after embolization** | systolic pressure (mmHg) | 73.3 | 15 |
| | diastolic pressure (mmHg) | 67.1 | 12.5 |
| **ANEURYSM VOLUME** | | | |
| | aneurysm volume ($mm^3$) | 371.1 | 840.7 |
| **STENT** | | | |
| + | packing (%) | 26.4 | 19.8 |
| | coil volume ($mm^3$) | 38.3 | 30.3 |
| - | packing (%) | 12.2 | 12.2 |
| | coil volume ($mm^3$) | 46.1 | 91.1 |

In 32/45 cases, coil embolization was assisted with a stent, for details please refer to Fig 2. Aneurysms embolized with stent assistance were significantly more densely coiled, compared to those where stent was not used–p = 0.015, yet the total coil volume did not differ significantly–p = 0.10 –Fig 3. Accordingly, Spearman Rank Correlation Coefficient showed a significant association between the presence of a stent and packing density–r = 0.36, p<0.05.

Presence of the stent showed significant reverse correlation with intra-aneurysmal pressure–both systolic and diastolic—after its implantation (r = -0.70 and r = -0.75, respectively), which was further supported by correlations with stent cell size–r = 0.72 and r = 0.71, respectively (P<0.05). Stent implantation resulted in significant decrease in diastolic intra-aneurysmal pressure (p = 0.046) (Fig 4). Systolic or mean intra-aneurysmal pressure did not differ significantly.

**Table 2. Descriptive statistics of treated aneurysms.**

| LOCATION | DMAX (mm) | WIDTH (mm) | HEIGHT (mm) | NECK (mm) | DELTA ANGLE (°) | STENTING (%) |
|---|---|---|---|---|---|---|
| ICA | 6.8±4.2 | 7.8±3.9 | 5.9±3.7 | 5.3±1.7 | 112.8±32.4 | 77 |
| MCA | 13.9±14.1 | 9.8±6.9 | 12.5±14.2 | 3.7±1.8 | 140.5±36.1 | 0 |
| BA | 6.6±2.9 | 6.4±3.4 | 5.9±2.9 | 5.5±3.1 | 139.3±37.3 | 83 |
| ACA | 3.4 ±0.4 | 4.3±0.8 | 2.9±0.4 | 3.1±0.7 | 156.8±36.8 | 33 |

ICA–internal carotid artery, MCA–middle cerebral artery, BA–basilar artery, ACA–anterior cerebral artery, Dmax–length of aneurysm sac, delta angle–angle formed between the axis of proximal part of parent artery and Dmax

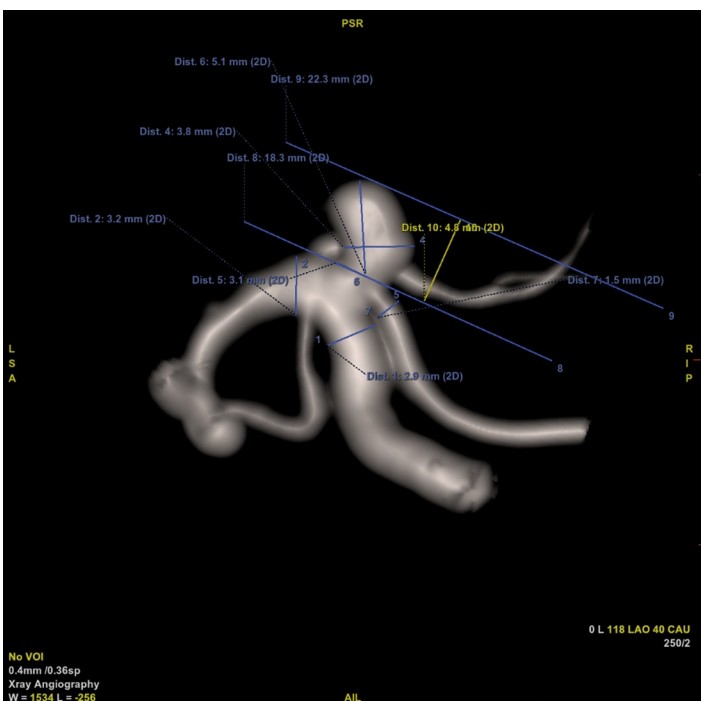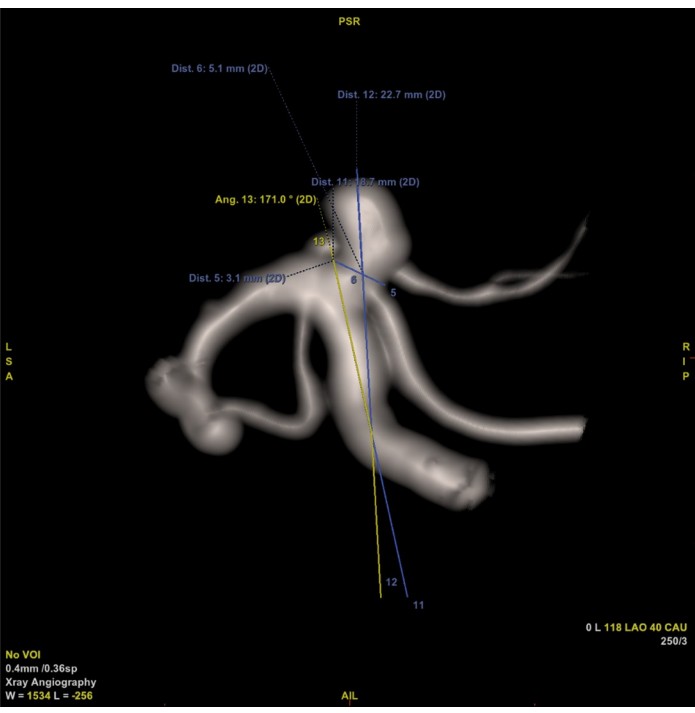

**Fig 1. Schematic morphological measurements on 3D angiography.** Please refer to caption for Table 2. Dist. 1 –diameter of afferent parent artery; Dist. 2 –diameter of efferent parent artery; Dist. 4 –width of the sac; Dist. 6 –length of the sac (Dmax); Dist. 10 –height of the sac; Ang. 13 –delta angle.

Embolization did not significantly change the intra-aneurysmal pressure in matched pairs, regardless of the use of stent. In cases of stent-assisted coiling p values were 0.69, 0.81, and 0.91 for systolic, diastolic and mean intra-aneurysmal pressure; respectively. In a subgroup with no stenting, the respective p values were 0.31, 0.21, and 0.21. Coil packing density was not significantly associated with intra-aneurysmal pressure after embolization

The effect on intra-aneurysmal pressures between stents was analyzed. Both non-normalized and normalized measurements were compared (respective to systemic values as a ratio to account for potential variations in systemic parameters). Values were presented Table 3.

Among aneurysm morphology characteristics, significant association was confirmed between the diameter of the aneurysm neck and the need to use a stent–r = 0.32, p<0.05. Similarly, the presence of more than two arteries originating around the aneurysm neck rendered the use of a stent necessary–r = 0.43, p<0.05. None of the measured intra-saccular pressures were significantly correlated with aneurysm morphology (p>0.05).

## Discussion

Intra-aneurysmal pressure would theoretically change in two situations: (1) replacing the blood in the aneurysm sac with embolization material, (2) changing the amount of blood flowing into the sac from the parent vessel. Both pathways were investigated in our study.

Endovascular coiling aims at preventing blood flow into the aneurysm by partially filling the aneurysm sac. Even though, coils alone cannot fill the entire aneurysm, they provoke intra-aneurysmal thrombosis, which eventually leads to complete occlusion. There are mixed data on the effect of aneurysm embolization on intra-saccular pressure. We found embolization did not significantly change the intra-aneurysmal pressure in matched pairs, regardless of the use of stent, which is in accordance with previous experimental and numerical-based research

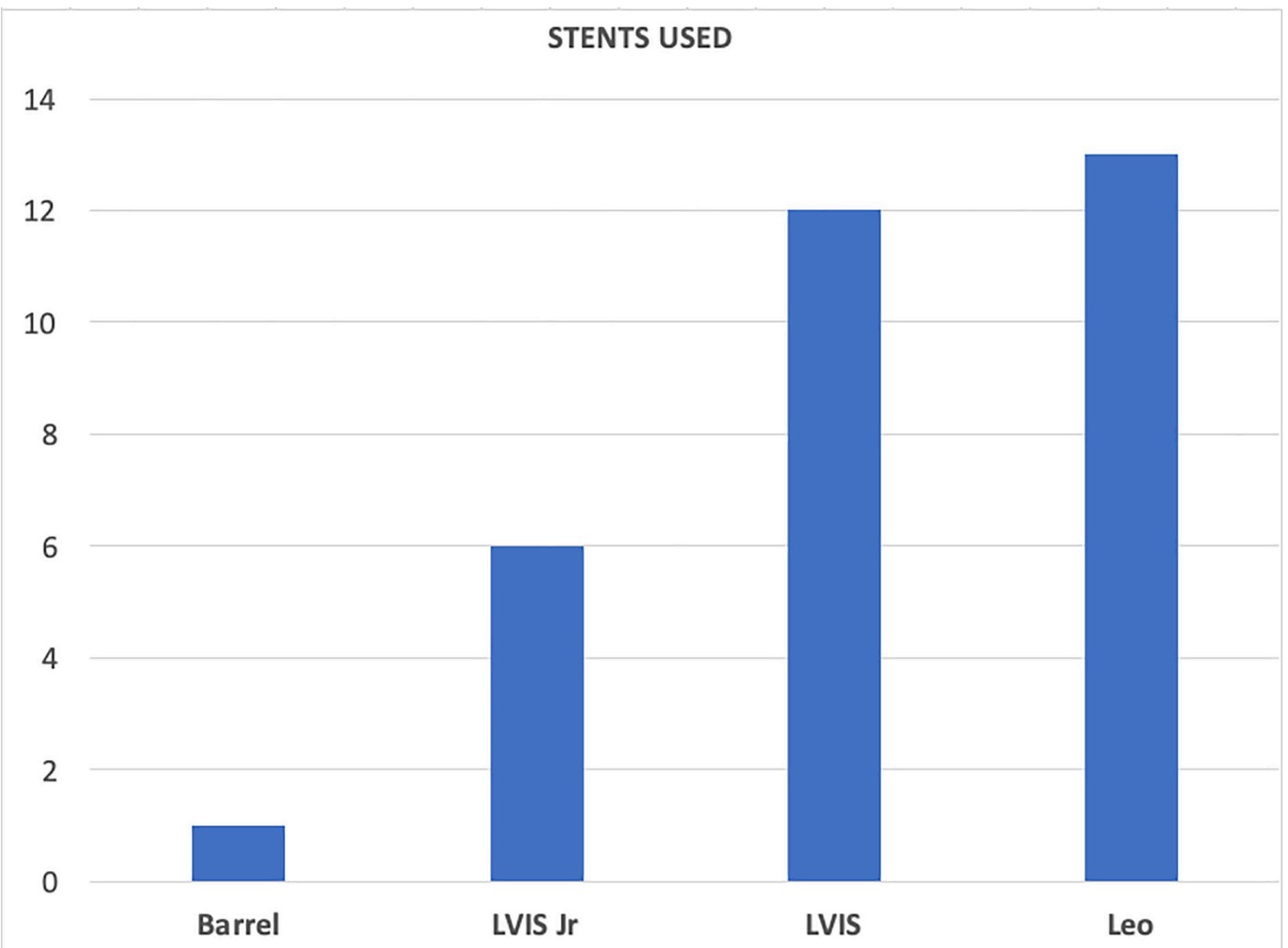

**Fig 2. Number of devices used for stent-assisted coiling.**

[13–16]. Coil packing density showed no significant association with post-embolization pressure measurements, even though coil packing density averaged 23%, with 3 cases of over 60%. Similarly, Groden et al. found no change in the pressure inside the aneurysm sac after up to 20% of coil packing [15]. In vitro observations led to similar conclusions where packing densities reached 93% [14]. Most of in-vivo-based published research focused on the differences between systemic, parent artery and intra-saccular pressure alone [17–20].

Despite the angiographic success of coiling, the flow in the aneurysm sac remains not affected enough to render significant pressure change. In about one-third of cases, coiling does not lead to adequate thrombus formation; these emphasize the role of hemodynamics in the process. As a consequence, recanalization and aneurysm recurrence can ensue. The potential is especially visible in wide-neck and large aneurysms [21]. Blood stagnation promoting thrombus formation in the aneurysm sac could be a result of decreased velocity of blood flow. Its reduction depends on coil packing density; when the packing density is low, coil configuration plays an important role [16, 22]. Coil orientation and packing density at the neck are directly responsible for flow velocity inside the aneurysm [23]. Effective packing density could

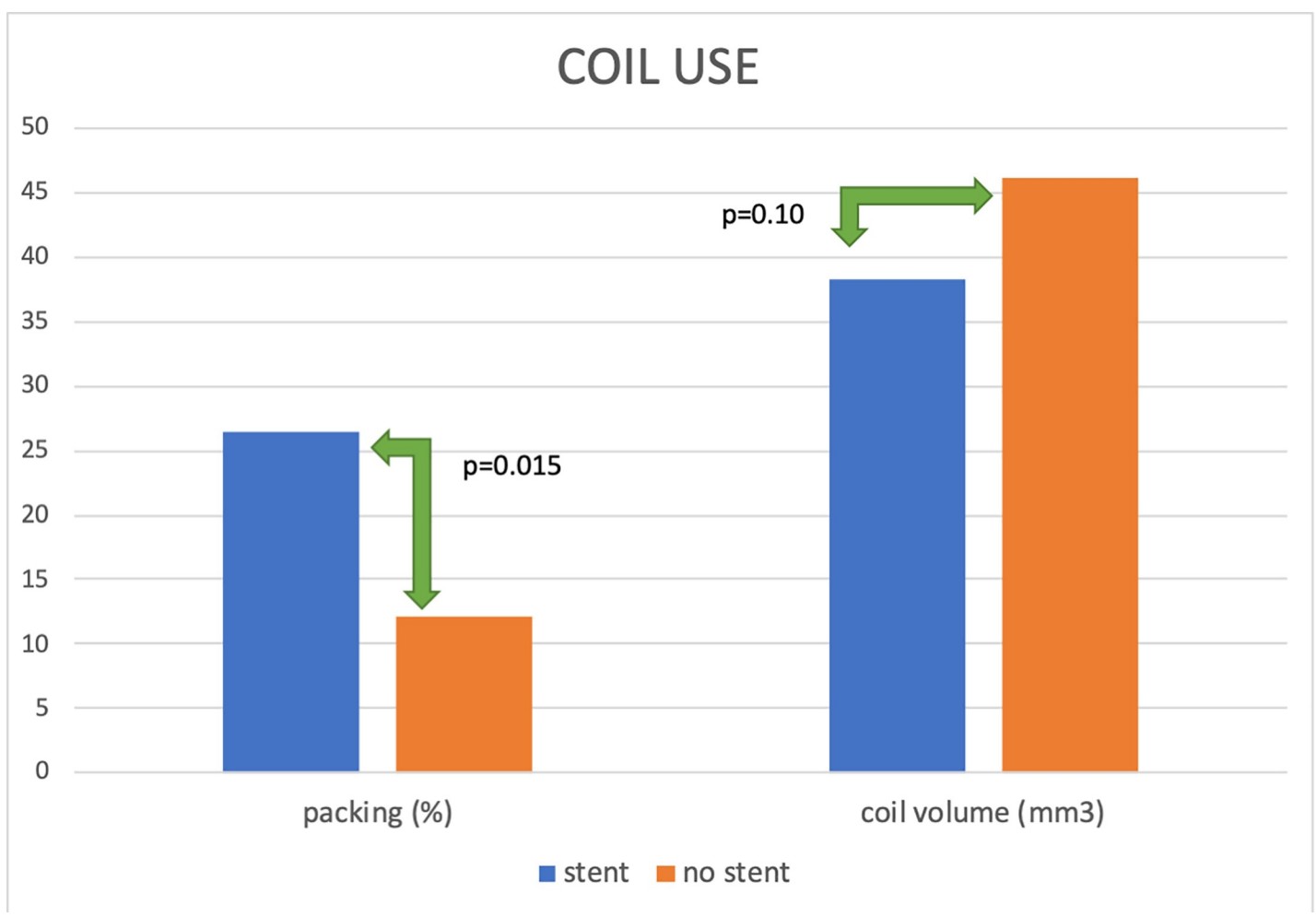

**Fig 3. Differences in coil use between stent-assisted and not assisted embolization.**

be especially challenging in case of the wide neck aneurysm, when there is a risk of coil hernia-tion into the parent vessel. Stents can stabilize the position of the coils. Indeed, we found coil packing density significantly increased when the stent was utilized. It could be explained by stent struts facilitating a more stable position of the microcatheter inside the aneurysm sac.

Stents themselves could promote aneurysm thrombosis by facilitating endothelialization of the aneurysmal orifice and diverting the blood flow away from the aneurysm [24]. These effects are considered to be related to stent porosity. In vitro studies indicate that high porosity —laser-cut—stents tend to have minimal effects on intra-aneurysmal flow [25]. Seshadhri et al. studied different wire densities and showed the flow-diverter with the highest wire den-sity induced the most significant hemodynamic change [26].

There are inconsistent reports on the relationship of intra-aneurysmal pressure after stent implantation, done predominantly on flow-diverters [26–29]. Paper published by Corriveau et al. has shown in in-vivo measurements that flow-diverter stent implantation results in an increase of the intra-saccular pressure inside large or giant intracranial aneurysm [30]. They theorized the phenomenon is the result of outflow obstruction, which in consequence could lead to delayed saccular rupture in angiographically successful cases. Schneiders et al. per-formed in-vivo measurements before and after flow-diverter placement, in a giant aneurysm;

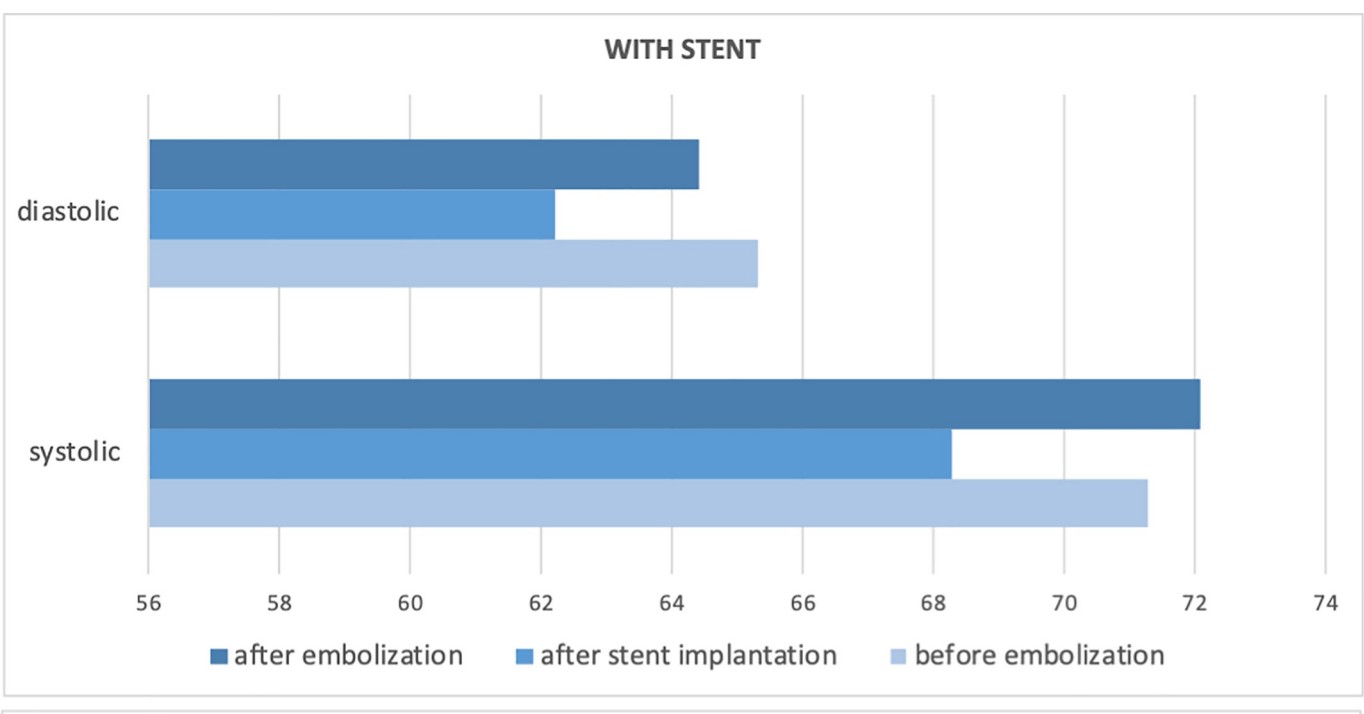

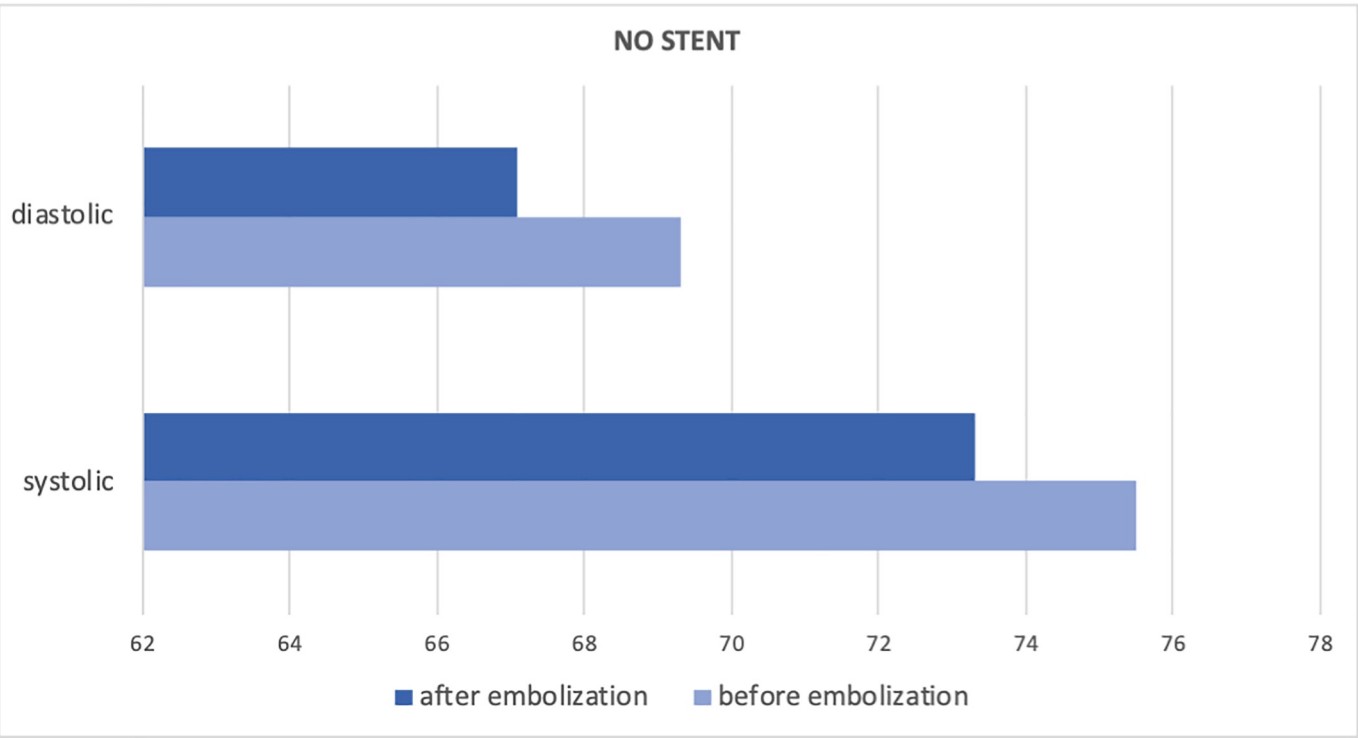

**Fig 4. Differences in systolic and diastolic intra-aneurysmal pressure between stent-assisted and not assisted embolization.**

they found transient decrease in intra-saccular pressure, which was restored to original within minutes [31]. All stents in this study, except one, were braided self-expanding closed-cell stents–Leo, LVIS and LVIS Jr. Each of them has characteristic cell size, innate to its design—0.9mm, 1mm, and 1.5mm; respectively. Although they were not originally intended for use as

Table 3. Comparison of intra-aneurysmal pressure after stent implantation and after embolization between different devices.

| POSITION | VARIABLE | LVIS vs LEO | LVIS vs LVIS Jr | LEO vs LVIS Jr |
|---|---|---|---|---|
| **aneurysm dome after stent implantation** | non-normalized systolic pressure | 0.15 | 0.22 | 0.007 |
| | non-normalized diastolic pressure | 0.33 | 0.15 | 0.02 |
| | normalized mean pressure | 0.97 | 0.77 | 0.72 |
| **aneurysm dome after embolization** | non-normalized systolic pressure | 0.36 | 0.81 | 0.64 |
| | non-normalized diastolic pressure | 0.91 | 0.73 | 0.76 |
| | normalized mean pressure | 0.48 | 0.62 | 0.72 |

Values represent p-values.

a flow-diverter they have shown potential to occlude small aneurysms in monotherapy, especially when telescoping stenting technique was used [32, 33]. We showed the implantation of the stent leads to significant decrease in diastolic pressure within the aneurysm sac (p = 0.046). Systolic or mean intra-aneurysmal pressure did not differ significantly. Given the significant reverse correlation of stenting with intra-aneurysmal pressure, further supported by similar relationship with stent cell size, we decided to compare the effect between stents. The differences in intra-saccular pressure after stent placement reached significance, when Leo and LVIS Jr devices were compared. These observations were not reproduced when aneurysms were coiled. It could be attributable to non-uniform coil packing density between the aneurysms.

In view of all observations, any significant change of intra-saccular pressure may lead to delayed aneurysm rupture by: (a) outflow obstruction–higher pressure, or (b) generating local drop of wall shear stress (WSS)–i.e. stasis at the periphery of the sac, and secondary inflammation–lower pressure. Taking clinical perspective into consideration, it might be reasonable to promote uniform stasis throughout the aneurysmal sac, by coiling the aneurysm, regardless of type of stent–flow-diverter or non-flow-diverter.

Although in our study series, we found no association between morphological parameters of the aneurysm sac and pressure measurements, CFD simulations found aneurysm and parent artery geometry to be relevant to their hemodynamics [34–36]. Aneurysm volume, sac depth, neck maximum width and neck area showed inverse relationship with wall shear stress, i.e. increase in the former resulted in decrease of the latter. Decrease in WSS promoted local blood stasis, as mentioned already, which might lead to rupture of the aneurysm–innate due to large aneurysm sac, or secondary due to suboptimal disruption of the flow inside the sac. The hemodynamic effect of stenting was mostly affected by aneurysm morphology, less by its position or orientation relative to parent vessel [34]. Meng et al. demonstrated the higher the parent vessel curvature the lower effect does stenting have on intra-aneurysmal flow reduction [35]. Impact of stents on inflow rate and mean velocity is more effective in narrow-necked cerebral aneurysms [36]. In view of our data, combined with the latest reports, there is added value to the utility of low-profile braided stents in proximal segments of cerebral vasculature. Not only do the struts support the coils, which could be more densely packed, but also impede the inflow of blood into the aneurysm sac–especially in lower curvature or side-wall aneurysms.

Our study has limitation inherent to single-center cohort. Elective treatment of non-ruptured aneurysms was performed. Location and morphology of the aneurysm were not uniform, although special care was taken to center the microcatheter position for all pressure readings. Measurements taken through long microcatheter bear the risk of substantial impedance and destructive interference effect on the accuracy of measurements, yet these were independently validated in published reports [17–20].

Finally, thrombus formation does not depend on hemodynamics and embolic materials alone. The process is complex, dependent also on a variety of hematologic factors such rheological properties and platelet function (medication-driven included) [27, 29, 37]. Substantial reduction of flow might limit the supply of prothrombotic factors to secure stable thrombus formation.

In conclusion, low-profile braided stents show a potential to divert blood flow–a property shown in our study to be related to their porosity–as reflected in significant decrease in diastolic pressure after stent placement. Coiling does not significantly change the intra-aneurysmal pressure, regardless of packing density.

## Supporting information

**S1 File.**
(PDF)

## Author Contributions

**Conceptualization:** Piotr Piasecki, Jerzy Narloch.

**Data curation:** Piotr Piasecki, Piotr Ziecina, Krzysztof Brzozowski, Marek Wierzbicki, Jerzy Narloch.

**Formal analysis:** Piotr Piasecki, Krzysztof Brzozowski, Jerzy Narloch.

**Investigation:** Piotr Piasecki, Piotr Ziecina, Marek Wierzbicki.

**Methodology:** Piotr Piasecki, Piotr Ziecina, Krzysztof Brzozowski, Marek Wierzbicki, Jerzy Narloch.

**Software:** Piotr Ziecina, Krzysztof Brzozowski, Marek Wierzbicki, Jerzy Narloch.

**Supervision:** Jerzy Narloch.

**Validation:** Piotr Piasecki, Krzysztof Brzozowski.

**Visualization:** Piotr Piasecki.

**Writing – original draft:** Piotr Piasecki, Jerzy Narloch.

**Writing – review & editing:** Piotr Piasecki, Jerzy Narloch.

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
