## [Decision Letter · Decision Letter 0]

24 Apr 2020

PONE-D-20-00340

Intra-aneurysmal pressure changes during stent-assisted coiling

PLOS ONE

Dear Dr Narloch,

Thank you for submitting your manuscript to PLOS ONE. After careful consideration, we feel that it has merit but does not fully meet PLOS ONE’s publication criteria as it currently stands. Therefore, we invite you to submit a revised version of the manuscript that addresses the points raised during the review process.

The article is interesting but it needs to be revised according to reviewers' suggestions.

We would appreciate receiving your revised manuscript by Jun 08 2020 11:59PM. To enhance the reproducibility of your results, we recommend that if applicable you deposit your laboratory protocols in protocols.io, where a protocol can be assigned its own identifier (DOI) such that it can be cited independently in the future. For instructions see: http://journals.plos.org/plosone/s/submission-guidelines#loc-laboratory-protocols

We look forward to receiving your revised manuscript.

Kind regards,

Prof. Raffaele Serra, M.D., Ph.D

Academic Editor

PLOS ONE

Journal Requirements:

2. Please amend either the abstract on the online submission form (via Edit Submission) or the abstract in the manuscript so that they are identical.

3. Thank you for including your ethics statement:  "Written informed consent obtained was obtained. The study was approved by Institutional Review Board (decision 43/WIM/2011)."

Additional Editor Comments (if provided):

The manuscript is potentially interesting and will be reconsidered provided the authors are willing to amend the manuscript according to reviewers' concerns.

Reviewers' comments:

Reviewer's Responses to Questions

**Comments to the Author**

1. Is the manuscript technically sound, and do the data support the conclusions?

Reviewer #1: Yes

Reviewer #2: Yes

2. Has the statistical analysis been performed appropriately and rigorously? 

Reviewer #1: Yes

Reviewer #2: Yes

3. Have the authors made all data underlying the findings in their manuscript fully available?

Reviewer #1: Yes

Reviewer #2: Yes

4. Is the manuscript presented in an intelligible fashion and written in standard English?

Reviewer #1: Yes

Reviewer #2: Yes

5. Review Comments to the Author

Reviewer #1: The authors aimed to examine aneurysm hemodynamics with intra-saccular pressure measurement, and compare the effects of coiling, stenting and stent-assisted coiling in proximal segments of intracranial circulation. The manuscript deserves consideration. The Discussion must be improved a little bit. You should discuss more widely, expanding this section, on the importance of your findings in the light of the current literature, and highlighting the clinical effects of your results.

Reviewer #2: I really appreciated this manuscript. There are good insight of physics and hemodynamics. Nevertheless, I think that one or two diagrams (provided as images) may help readers to better understand your findings.

6. PLOS authors have the option to publish the peer review history of their article (what does this mean?). If published, this will include your full peer review and any attached files.

Reviewer #1: No

Reviewer #2: No

---

## [Author Response · Author response to Decision Letter 0]

29 Apr 2020

Dear Editors and Reviewers,

Thank you for your letter and for the reviewers’ comments concerning our manuscript “Intra-aneurysmal pressure changes during stent-assisted coiling.”

Thank you for the valuable reviewer’s comments. The critique was very constructive and relevant. The comments provided significant guidance to our study. We have carefully studied your comments and have made the recommended corrections and revisions. We hope that our revised manuscript is acceptable after the revision.

Please find below our corrections, and the responses to the reviewer’s comments are as

follows:

Reviewer #1: “The authors aimed to examine aneurysm hemodynamics with intra-saccular pressure measurement, and compare the effects of coiling, stenting and stent-assisted coiling in proximal segments of intracranial circulation. The manuscript deserves consideration. The Discussion must be improved a little bit. You should discuss more widely, expanding this section, on the importance of your findings in the light of the current literature, and highlighting the clinical effects of your results.”

Our response: We expanded the discussion to highlight the clinical effect of our observations. Hopefully, it has improved. Please refer to lines: 232-243, 254-256, and 262-266.

Reviewer #2: “I really appreciated this manuscript. There are good insight of physics and hemodynamics. Nevertheless, I think that one or two diagrams (provided as images) may help readers to better understand your findings”.

Our response: We appreciate kind comments. Necessary changes were made. Additional figures were introduced – fig. 3 and 4.

Kind regards,

Authors

---

## [Decision Letter · Decision Letter 1]

18 May 2020

Intra-aneurysmal pressure changes during stent-assisted coiling

PONE-D-20-00340R1

Dear Dr. Narloch,

We are pleased to inform you that your manuscript has been judged scientifically suitable for publication and will be formally accepted for publication once it complies with all outstanding technical requirements.

With kind regards,

Prof. Raffaele Serra, M.D., Ph.D

Academic Editor

PLOS ONE

Additional Editor Comments (optional):

amended manuscript is acceptable

Reviewers' comments:

Reviewer's Responses to Questions

**Comments to the Author**

1. If the authors have adequately addressed your comments raised in a previous round of review and you feel that this manuscript is now acceptable for publication, you may indicate that here to bypass the “Comments to the Author” section, enter your conflict of interest statement in the “Confidential to Editor” section, and submit your "Accept" recommendation.

Reviewer #2: All comments have been addressed

2. Is the manuscript technically sound, and do the data support the conclusions?

Reviewer #2: Yes

3. Has the statistical analysis been performed appropriately and rigorously? 

Reviewer #2: Yes

4. Have the authors made all data underlying the findings in their manuscript fully available?

Reviewer #2: Yes

5. Is the manuscript presented in an intelligible fashion and written in standard English?

Reviewer #2: Yes

6. Review Comments to the Author

Reviewer #2: All my concerns have been addressed. The paper now can be accepted in the current format. Congratulations.

7. PLOS authors have the option to publish the peer review history of their article (what does this mean?). If published, this will include your full peer review and any attached files.

Reviewer #2: No

---

## [Editor Report · Acceptance letter]

22 May 2020

PONE-D-20-00340R1 

Intra-aneurysmal pressure changes during stent-assisted coiling 

Dear Dr. Narloch:

I am pleased to inform you that your manuscript has been deemed suitable for publication in PLOS ONE. Congratulations! Your manuscript is now with our production department. 

With kind regards,

on behalf of

Prof. Raffaele Serra 

Academic Editor

PLOS ONE